**Data Availability Statement:** All DPC datasets have ethical or legal restrictions for public deposition due to inclusion of sensitive information

# Dementia and patient outcomes after hip surgery in older patients: A retrospective observational study using nationwide administrative data in Japan

**Noriko Morioka** [1]*, **Mutsuko Moriwaki**[2], **Jun Tomio**[3], **Kiyohide Fushimi**[4], **Yasuko Ogata**[1]

1 Graduate School of Health Care Sciences, Tokyo Medical and Dental University, Tokyo, Japan, 2 Department of Tokyo Metropolitan Health Policy Advisement, Graduate School of Medicine, Tokyo Medical and Dental University, Tokyo, Japan, 3 Department of Public Health, Graduate School of Medicine, The University of Tokyo, Tokyo, Japan, 4 Health Policy and Informatics, Graduate School of Medicine, Tokyo Medical and Dental University, Tokyo, Japan

* morioka.gh@tmd.ac.jp

## Abstract

### Objective

To investigate whether dementia is associated with incidence of adverse events and longer hospital stays in older adults who underwent hip surgery, after adjusting for individual social and nursing care environment.

### Design and setting

Retrospective observational study using the linked data between the Japanese Diagnosis Procedure Combination database and the reports of the medical functions of hospital beds database in Japan (April 2016—March 2017).

### Participants

A total of 48,797 individuals aged 65 and older who underwent hip surgery and were discharged during the study period.

### Methods

Outcomes included in-hospital death, in-hospital pneumonia, in-hospital fracture, and longer hospital stay. We performed two-level, multilevel models adjusting for individual and hospital characteristics.

### Results

Among all participants, 20,638 individuals (42.3%) had dementia. The incidence of adverse events for those with and without dementia included in-hospital death: 2.11% and 1.11%, in-hospital pneumonia: 0.15% and 0.07%, and in-hospital fracture: 3.76% and 3.05%, respectively. The median (inter quartile range) length of hospital stay for those with and without

from the human subjects. All inquiries should be addressed to the ethics committee of Tokyo Medical and Dental University via e-mail: syomu1.adm@tmd.ac.jp.

**Funding:** This work was supported by a Grant-in-Aid for Scientific Research by the Ministry of Education, Culture, Sports, Science and Technology, Japan (No.18K17633); the grant by the Institute for Health Economics and Policy, Japan; and, a Grant-in-Aid for Research on Policy Planning and Evaluation from the Ministry of Health, Labour and Welfare, Japan (H30-Seisaku-Shitei-004). The funding sources played no role in the design and conduct of the study; collection, management, analysis, and interpretation of the data; preparation, review, or approval of the manuscript; or decision to submit the manuscript for publication.

**Competing interests:** The authors have declared that no competing interests exist.

dementia were 26 (19–39) and 25 (19–37) days, respectively. Overall, the odds ratios (95% confidence interval (CI)) of dementia for in-hospital death, in-hospital pneumonia, and in-hospital fracture were 1.12 (0.95–1.33), 0.95 (0.51–1.80), and 1.08 (0.92–1.25), respectively. Dementia was not associated with the length of hospital stay (% change) (-0.7%, 95% CI -1.6–0.3%). Admission from home, discharge to home, and lower nurse staffing were associated with prolonged hospital stays.

## Conclusions

Although adverse events are more likely to occur in older adults with dementia than in those without dementia after hip surgery, we found no evidence of an association between dementia and adverse events or the length of hospital stay after adjusting for individual social and nursing care environment.

## Introduction

Hip fracture is a major injury among older adults that can lead to substantial loss of healthy life-years [1]. Older adults with dementia are also more likely to incur hip fractures than those without dementia [2–4]. Due to the rapidly increasing prevalence of dementia in the general population [5], its occurrence among patients with hip fractures has also increased in acute care settings. Systematic reviews suggested that about 20–50% of the individuals admitted to acute care hospitals due to hip fractures have dementia or a cognitive impairment [6, 7]. Among people who underwent hip surgery, retrospective cohort studies investigated the association between dementia and patient outcomes, in Australia, Canada, the Netherlands, and Japan [8–12]. Compared to those without dementia, people with dementia were more likely to have a higher delirium rate [8], more post-surgical complications [12], be admitted to long-term care [13], and be readmitted within 30 days after discharge [9].

However, reports on the association between dementia and the length of hospital stay after hip surgery are inconsistent. A shorter hospital stay among people with dementia than in those without dementia has been indicated [8, 10]; however, this association varied depending on patients' social factors such as type of residence before the hip fracture—a longer hospital stay was reported for community-dwelling people than for long-term care facility users [13]. A better understanding of the association between dementia and patient outcomes after hip surgery, considering residence type before and after hospitalization, will help promote integrated care for people with dementia in aging societies.

Moreover, providing adequate care for individuals with dementia in hospital settings has been challenging [14, 15]. Nursing care environment factors such as nurse staffing level, skill mix, and advanced skill for dementia and gerontological nursing care are essential for providing adequate care to people with dementia [16]. However, few studies have considered the effect of the nursing care environment when investigating the association between dementia and patient outcomes [14, 17]. We, therefore, aimed to investigate the associations between dementia, adverse events, and length of hospital stay among older adults who underwent hip surgery considering the participants' social and nursing care environment.

## Methods

### Study design

We conducted a retrospective observational study using nationwide data in Japan.

## Data source

We used the Japanese Diagnosis Procedure Combination database and reports of the medical functions of hospital beds in the Medical Care Act via hospital record number. The aforementioned database contains nationwide data of acute care in-patients, comprising administrative claims data and discharge information of approximately 5 million discharged patients (in 2016) from 1,667 acute care hospitals in Japan, which represents approximately 50% of all acute care in-patient hospitalizations in the country.

The database includes patients' demographic information (i.e., age, sex, height, weight), activities of daily living (ADL) scores at admission and at discharge, patients' place of residence before admission and after hospitalization, main diagnoses, pre-existing comorbidities, and post-admission complications as per the International Statistical Classification of Disease and Related Health Problems 10th Revision (ICD-10) codes, and surgical procedures as per the Japanese original surgical coding system. Additional details of the database are provided elsewhere [18, 19]. In this study, we could not include ADL data in the analysis because almost one-fourth of such data were missing.

Reports of the medical functions of hospital beds include ward characteristics (e.g., the total number of in-patients during the past year), the number of full-time nurses and associate nurses, and the number of beds among others. The Medical Care Act [20] mandates all hospitals with general or long-term care beds to report their status to prefectural governors every year to build an appropriate regional medical care system.

## Participant selection

We selected individuals aged 65 or older during hospital admission and underwent hip surgery, osteosynthesis, bipolar hip arthroplasty, or total hip arthroplasty at acute care hospitals (average number of in-patients per day was $\geq$ 200) from April 2016 to March 2017. Exclusion criteria were: unconscious at admission, which was assessed using the Japan Coma Scale (i.e., level II or worse; that is, incapable of opening eyes with stimulation) [21] and death within 24 hours of admission.

## Outcomes

We used in-hospital mortality, in-hospital pneumonia, in-hospital fracture, and length of hospital stay as the health outcome variables. In-hospital mortality was defined as all-cause death during hospitalization. In-hospital pneumonia was identified by the type of pneumonia; in the database, pneumonia severity is classified into three categories: community-acquired, in-hospital, and other. In-hospital fracture was defined as at least one fracture post-admission: skull and facial bones fracture (S02), neck fracture (S12), fracture of rib(s), sternum, and thoracic spine (S22), fracture of lumbar spine and pelvis (S32), fracture of shoulder and upper arm (S42), forearm fracture (S52), wrist and hand level fracture (S62), femur fracture (S72), fracture of lower leg including ankle (S82), fracture of foot excluding ankle (S92), fractures in multiple body regions (T02), spine fracture (T08), upper-limb fracture (T10), lower limb fracture (T12), and fracture of unspecified body region (T142). Length of hospital stay was defined as the number of hospitalization days from admission to discharge. When calculating length of hospital stay, in-hospital death cases were excluded.

## Dementia status

Dementia status was the primary explanatory variable. For dementia, the sensitivity and specificity in our database were 37.5% and 100%, respectively [22]; therefore, we used multiple

definitions in addition to the diagnosis, in line with previous studies [9, 23]. Dementia was defined as the presence of at least one prescription of an antidementia drug (e.g., donepezil, galantamine, memantine, rivastigmine) during hospitalization, a diagnosis of dementia at admission (ICD-10 codes: F00–F03, F05.1, G30, and G31), or the level of "ADLs of people with dementia" being I or greater at hospital admission. "ADLs of people with dementia" is the scale used for the assessment of dementia in the long-term care insurance schema in Japan. In the database, the scale was classified into three categories: 0 ("without dementia"), I–II ("individuals with dementia who live independently with assistance"), and III–IV or M ("individuals with dementia who require moderate or full-time care").

## Individual characteristics

We selected covariates as potential confounders from previous studies. We obtained data regarding demographic characteristics (age and sex), clinical characteristics (body mass index [BMI], and comorbidities), type of surgery (osteosynthesis, bipolar hip arthroplasty, or total hip arthroplasty), and place of residence before admission (home, long-term care facilities, and others including another hospital). Comorbidity was classified using the modified form of the Charlson comorbidity index (CCI) [24]. A CCI score of 0, 1, 2, or $\geq$ 3 at admission was calculated based on ICD-10 diagnoses for comorbidities. Psychotropic drug use was defined in at least one prescription for hypnotic, anxiolytic, or psychotropic medications during hospitalization.

## Nursing staffing level, expertise for dementia care, and hospital size

In Japan, nurses and associate nurses are qualified under the Act on Public Health Nurses [25]. The former is a national license and the latter is a license by the prefectural government; however, both are effective anywhere in Japan. To measure nurse staffing levels, we used the patient-to-nurse ratio and skill mix which have been widely used in previous studies [26, 27]. The average number of in-patients per nurse and associate nurses per shift at general acute care wards were calculated using the following equation:

*The average number of in-patients per nurse per shift = (the number of in-patients in the general care wards during the past year\* 3 \* 8) / (the number of full-time equivalent nurses working at general wards/1800)*

Japan's Ministry of Health, Labour and Welfare defined the annual working hours of full-time equivalent nurses as 1,800 hours (three shifts a day; 8 hours each) when calculating the patient-to-nurse ratio per shift [28]. A lower number indicates a lower workload for nurses and better nurse staffing. As an indicator of skill mix, we used the percentage of nurses among the total nursing staff (nurse and associate nurse) in general acute care wards.

Special dementia care status was determined by the status of financial incentives for the same. In Japan, the financial incentives for special dementia care in the fee schedule were introduced in April 2016: Type 1 comprises a dementia-specific multidisciplinary team consisting of physicians skilled at dementia management, advanced practice nurses for dementia or gerontological nursing, and psychiatric social workers, and Type 2 comprises two or more nurses who have been trained regarding assessment and care for dementia in general wards [23, 29]. We addressed this as a categorical variable (e.g., none, Type 1, or Type 2). We also considered the hospital size (number of beds) and type of established organization (national; public; social; private; and others including public interest corporations, private university

corporations, social welfare corporations, medical co-op, companies, and other corporations) as other hospital characteristics.

## Statistical analyses

We removed all cases with missing data in any of the selected covariates and conducted chi-square tests for categorical data and t-tests or Mann-Whitney U test for numerical data to compare patients with and without dementia. To investigate the association between dementia and the length of hospital stay, after adjusting for covariates and taking the correlated structure of the data into account, we conducted a two-level random intercept model with patients nested within hospitals [30]. We took logarithm of length of hospital stay in the regression model because the distribution of length of hospital stay was skewed. In this "semi-logarithmic" model transformation regressed log y on X, the interpretation of regression coefficients was approximately the percentage change in y resulting from a one-unit change in X. The exact percentage change, $\%\Delta y = 100 * [\exp (b_j \Delta X_j)-1]$, would give us a more accurate prediction of the change in length of hospital stay [31]. We also conducted two-level, multilevel logistic models to estimate odds ratios (ORs) of dementia for the other dichotomous outcomes (in-hospital pneumonia, in-hospital fracture, and in-hospital mortality) adjusting for the covariates. In all models, the intraclass correlation coefficient of the null models with no explanatory variables $\geqq 0.1$ suggested appropriateness of the use of multilevel analysis [30]. All analyses were performed with Stata version 15 (StataCorp, College Station, TX, USA).

## Ethical considerations

Study approval was obtained from the Institutional Review Board at the Tokyo Medical and Dental University (No.M2000-788). The need for informed consent was waived because of the anonymous nature of the data.

## Results

We obtained data from 48,049 eligible patients from 404 hospitals (S1 Fig). Of them, 20,638 (43.0%) had dementia (Table 1). The dementia group had more women, long-term care facility users before admission, and patients with higher CCI scores (Table 1) compared to those in the without dementia group.

Table 2 shows the characteristics of all the 404 hospitals. Most were public and had no special care for dementia. Moreover, most of the nursing staff comprised nurses and, in average, 6.3 patients were allocated per nurse and associate nurse.

The overall number (proportions) of in-hospital deaths, in-hospital pneumonia, and in-hospital fractures were 748 (1.53%), 51 (0.10%), and 1,634 (3.35%), respectively. The prevalence of in-hospital death, in-hospital pneumonia, and in-hospital fracture for people with and without dementia were 2.11% vs. 1.11% (p < .001), 0.15% vs. 0.07% (p = .007) and 3.76% vs. 3.05% (p < .001), respectively (Table 3). The medians (inter quartile range) length of hospital stay in the total sample and in people with and without dementia were 26 (19–38), 26 (19–39) and 25 (19–37) days, respectively (p = .758).

The adjusted ORs (95% CI) of dementia for in-hospital death, in-hospital pneumonia, and in-hospital fracture were 1.12 (0.95–1.33), 0.95 (0.51–1.80), and 1.08 (0.92–1.25), respectively (Table 4 and S1 Table).

Table 5 and S2 Table show that dementia was not significantly associated with the length of hospital stay after adjusting for individual and hospital characteristics (-0.7%, 95% confidence interval (CI) = -1.6–0.3%). Shorter hospital stays were associated with individuals' place of residence before admission and after discharge compared with from/to home, admission from

**Table 1. Characteristics of the study population with and without dementia.**

| | Total | | With dementia | | Without dementia | | P |
|---|---|---|---|---|---|---|---|
| | N = 48,797 | | n = 20,638 | | n = 28,159 | | |
| | n/mean | %/SD | n/mean | %/SD | n/mean | %/SD | |
| Female, n, % | 38,830 | 79.6 | 16,648 | 80.7 | 22,182 | 78.8 | <0.001 |
| Age, years, mean, SD | 82.2 | 8.3 | 86.1 | 6.7 | 79.3 | 8.1 | <0.001 |
| Body mass index, mean, SD | 21.2 | 3.8 | 20.3 | 3.5 | 21.9 | 3.9 | <0.001 |
| Charlson comorbidity index, n, % | | | | | | | <0.001 |
| 0 | 22,493 | 46.1 | 6,801 | 33.0 | 15,692 | 55.7 | |
| 1 | 15,289 | 31.3 | 8,006 | 38.8 | 7,283 | 25.9 | |
| 2 | 7,122 | 14.6 | 3,939 | 19.1 | 3,183 | 11.3 | |
| ≥ 3 | 3,893 | 8.0 | 1,892 | 9.2 | 2,001 | 7.1 | |
| Place of residence before admission, n, % | | | | | | | <0.001 |
| Home | 37,619 | 77.1 | 12,415 | 60.2 | 25,204 | 89.5 | |
| Long-term care facility | 7,654 | 15.7 | 6,323 | 30.6 | 1,331 | 4.7 | |
| Other (hospital, clinic, etc.) | 3,524 | 7.2 | 1,900 | 9.2 | 1,624 | 5.8 | |
| Place of residence after discharge, n, % | | | | | | | <0.001 |
| Home | 15,839 | 32.5 | 3,700 | 17.9 | 12,139 | 43.1 | |
| Long-term care facility | 6,741 | 13.8 | 5,572 | 27.0 | 1,169 | 4.2 | |
| Other (hospital, clinic, etc.) | 26,217 | 53.7 | 11,366 | 55.1 | 14,851 | 52.7 | |
| Psychotropic drug prescription, n, % | 32,207 | 66.0 | 14,776 | 71.6 | 17,431 | 61.9 | <0.001 |
| Type of surgery, n, % | | | | | | | |
| Osteosynthesis | 26,327 | 54.0 | 13,763 | 66.7 | 12,564 | 44.6 | <0.001 |
| Bipolar hip arthroplasty | 13,865 | 28.4 | 6,397 | 31.0 | 7,468 | 26.5 | <0.001 |
| Total hip arthroplasty | 8,743 | 17.9 | 540 | 2.6 | 8,203 | 29.1 | <0.001 |

SD, standard deviation. Chi-square test or t-test.

**Table 2. Hospital characteristics (N = 404).**

| | Mean/% | SD |
|---|---|---|
| Number of beds | 439.0 | 167.9 |
| Number of patients per nurse and associate nurse | 6.3 | 0.8 |
| Percentage of nurses among all nursing staff | 83.1 | 6.0 |
| | n | % |
| Types of established organization of hospitals | | |
| National | 41 | 10.2 |
| Public | 168 | 41.6 |
| Social | 20 | 5.0 |
| Private | 73 | 18.1 |
| Others[a] | 102 | 25.3 |
| Addition to special care for dementia in the fee schedule | | |
| None | 205 | 50.7 |
| Type 2 (trained nurses) | 99 | 24.5 |
| Type 1 (multidisciplinary dementia care team) | 100 | 24.8 |

SD, standard deviation.

[a] public interest corporations, private university corporations, social welfare corporations, medical co-op, companies, and other corporations.

**Table 3. Length of hospital stay and prevalence of outcomes in people with and without dementia.**

| | With dementia | Without dementia | P |
|---|---|---|---|
| | n = 20,638* | n = 28,159* | |
| In-hospital death, n (%) | 436 (2.11) | 312 (1.11) | < .001[a)] |
| In-hospital pneumonia, n (%) | 31 (0.15) | 20 (0.07) | .007[a)] |
| In-hospital fracture, n (%) | 776 (3.76) | 858 (3.05) | < .001[a)] |
| Length of hospital stay (days), median (IQR) | 26 (19–39) | 25 (19–37) | .758[b)] |

*For the length of hospital stay, 20,202 patients with dementia and 27,847 patients without dementia were included after excluding in-hospital death; IQR, inter quartile range

a), chi-square test

b), Mann-Whitney U test.

long-term care facilities (-20.4%, 95% CI = -21.5–-19.3%) and other places including hospitals (-10.6%, 95% CI = -12.0–-9.2%), and discharge to long-term care facilities (-9.8%, 95% CI = -11.3–-8.4%) and to other institutions including hospitals (-10.7%, 95% CI = -11.6–-9.8%). Regarding hospital nursing care environment, a higher number of in-patients per nurse and associate nurse was significantly associated with 7.8% (95% CI = 4.2–11.5%) longer length of hospital stay.

## Discussion

We conducted a retrospective observational study with a large sample using nationwide administrative data in Japan to explore the association between dementia and patient outcomes after hip surgery among older adults. Although in-hospital death, in-hospital pneumonia, and in-hospital fracture were more likely to occur in people with dementia than in those without dementia in the present study, they were not significantly associated with dementia after adjusting for potential confounders such as individual social factors, nurse staffing level, and dementia care expertise. The improvement of dementia care in acute care settings could explain our findings. A previous Japanese study using 2007–2010 data from the same database utilized in our research revealed that people with dementia are more likely to have post-surgical complications during hospitalization [12]. This discrepancy might be explained by the promotion of policies for dementia care in hospitals. In 2015, the Japanese government renewed its comprehensive strategy for the promotion of dementia care—the New Orange Plan—which aimed to create a friendly community for older adults with dementia [32]. This national strategy established the critical goal of providing appropriate medical care for patients with

**Table 4. Results of multilevel logistic regression analysis for in-hospital death, in-hospital pneumonia, and in-hospital fracture among people with and without dementia (n = 48,797).**

| | Univariate regression | | | | Multivariate regression | | | |
|---|---|---|---|---|---|---|---|---|
| | OR | 95% CI | | P | Adjusted OR | 95% CI | | P |
| In-hospital death | 1.88 | 1.62 | 2.19 | <0.001 | 1.12 | 0.95 | 1.33 | 0.181 |
| In-hospital pneumonia | 1.97 | 1.11 | 3.49 | 0.021 | 0.95 | 0.51 | 1.80 | 0.885 |
| In-hospital fracture | 1.37 | 1.20 | 1.57 | <0.001 | 1.08 | 0.92 | 1.25 | 0.344 |

Adjusted for sex, body mass index, Charlson comorbidity index, type of surgery, psychotropic drug use, residence of before admission, number of in-patients per nurse and associate nurse, percentage of nurses among all nursing staff, addition to special care for dementia in the fee schedule, types of established organization of hospitals and number of hospital beds.

CI, confidence interval; OR, odds ratio.

**Table 5. Results of multilevel analyses for length of hospital stay (n = 44,883).**

| | Coefficient | 95% CI | | P |
|---|---|---|---|---|
| With dementia (ref. without dementia) | -0.007 | -0.016 | 0.003 | 0.184 |
| Individual characteristics | | | | |
| The type of residence before admission (ref. home) | | | | |
| Long-term care facilities | -0.228 | -0.242 | -0.214 | <0.001 |
| Other (hospital, clinic, etc.) | -0.112 | -0.128 | -0.096 | <0.001 |
| The type of residence after discharge (ref. home) | | | | |
| Long-term care facilities | -0.103 | -0.119 | -0.087 | <0.001 |
| Other (hospital, clinic, etc.) | -0.113 | -0.124 | -0.103 | <0.001 |
| Hospital characteristics | | | | |
| Number of in-patients per nurse and associate nurse | 0.075 | 0.042 | 0.109 | <0.001 |
| Percentage of nurses among all nursing staff | 0.001 | -0.005 | 0.005 | 0.937 |
| Addition to special care for dementia in the fee schedule (ref. none) | | | | |
| Type 2 (trained nurses) | 0.045 | -0.020 | 0.111 | 0.175 |
| Type 1 (multidisciplinary dementia care team) | -0.020 | -0.085 | 0.045 | 0.542 |

Adjusted for sex, body mass index, Charlson comorbidity index, type of surgery, psychotropic drug use, types of established organization of hospitals and number of hospital beds. The exact percentage change, %$\Delta y$ = 100 * [exp (bj$\Delta$Xj)−1] gives a more accurate prediction of the change in length of hospital stay. CI, confidence interval.

dementia in acute care facilities. Following this, financial incentives for appropriate medical care for patients with dementia were introduced in April 2016, which are included in a revision concerning the reimbursement of medical fees [29]. Such policy changes in dementia care may have reduced the disadvantages faced by people with dementia in acute care facilities; however, further studies are necessary to investigate the effect of policy changes on health outcomes of the patients.

In addition, there is another reason why dementia was not associated with longer hospital stays in the present study. Participants' social factors may be key determinants of length of hospital stay in Japan rather than the dementia status. In the present study, people admitted from home or discharged to home were more likely to have longer hospitalizations than those admitted from and discharged to long-term care facilities and other hospitals, regardless of dementia status. Past research in other countries also reported that patients often quickly return to their previous nursing home and therefore have a shorter length of stay compared to those admitted from their own homes [8, 10].

In addition, Japan is well known for recording prolonged hospital stays than other countries in the Organization for Economic Co-operation and Development (OECD) [33]. The median length of hospital stay for people with and without dementia in the present study was 26 days; almost the same as the average length of hospital stay in hip fracture cases overall in acute care hospitals in Japan [34] but longer than those in Canada (no dementia = 17.5 days, dementia = 24.0 days [13]) or Australia (dementia < 15 days [10]). Currently, the government has promoted the transition of the elderly from hospitalization to welfare homes, which is expected to contribute to reduced length of hospital stay. Fee-schedule introduced financial incentive to encourage hospital discharges and use post-discharge protocols, fostering co-operation among hospitals, clinics, and long-term-care facilities, especially for patients with hip fractures [35]. In this context, the places of residence before admission and after discharge are key determinants on the decision of time of discharge for elderly people after hip surgery. People with dementia who were admitted from home need more time to prepare the adequate

post-hospitalization residence compared to those admitted from long-term care facilities [13]. Regarding hospital discharge, individuals going to another hospital and/or long-term care facility transition with no delays and seamlessly compared to those being discharged to their own homes, given that existing post-discharge protocols between acute and post-acute hospitals or long-term-care facilities work well [35].

Our study also contributed new findings concerning the factors related to older adult in-patients' health outcomes after hip surgery. Remarkably, our study suggested that not only individual characteristics but also the nursing staffing level is related to these outcomes. The increase of one in-patient per nursing staff was associated with longer length of hospital stay, in other words, increased nursing workload might disturb early discharge. This finding supported the results of Kane's meta-analysis, which concluded that an increase by one registered nurse per patient day was associated with a 31% shorter length of hospital stay in surgical patients (OR, 0.69; 95% CI, 0.55–0.86 [27]).

This study had several limitations. First, the data were obtained from relatively large hospitals in Japan. We selected hospitals that had an average of > 200 in-patients per day in order to adjust for the effect of surgeon procedure volumes, which are related to patient outcomes [36]. Our findings might thus not be generalizable. Second, there may be some unmeasured confounders that might affect patient outcomes (e.g., whether it was a scheduled or emergency surgery and the severity of dementia). Patients with severe dementia are more likely to choose non-operative treatment for hip fractures [37]. In this study, we focused on cases with hip surgery, which might have involved those with relatively mild dementia. The ADL of patients were also related to the length of hospital stay [8, 38]. Although our dataset included ADL, some ADL data were missing and thus had to be excluded from the analysis. Third, for the database used in this study, the validation study reported that the sensitivity of diagnoses was low and varied across conditions [22]. The prevalence of non-critical adverse events, such as mild pneumonia, might have been underestimated. This, however, would not affect our findings since underestimations could occur with or without dementia.

## Conclusion

Adverse events were more likely to occur in people aged 65 or older with dementia than in those without dementia after hip surgery. These events were not significantly associated with dementia after adjusting for individual social and nursing care environment. Further studies are necessary to identify factors that mitigate the effect of dementia on poor outcomes.

## Supporting information

**S1 Fig. Flowchart of sample selection.**
(PDF)

**S1 Table. Results of multivariate analyses for in-hospital death, in-hospital pneumonia, and in-hospital fracture and dementia (full model).**
(PDF)

**S2 Table. Results of multivariate analyses for length of hospital stay and dementia (full model).**
(PDF)

## Author Contributions

**Conceptualization:** Noriko Morioka, Jun Tomio.

**Data curation:** Mutsuko Moriwaki, Kiyohide Fushimi.

**Formal analysis:** Jun Tomio.

**Funding acquisition:** Noriko Morioka, Kiyohide Fushimi.

**Methodology:** Noriko Morioka, Jun Tomio.

**Project administration:** Noriko Morioka.

**Supervision:** Kiyohide Fushimi, Yasuko Ogata.

**Writing – original draft:** Noriko Morioka, Jun Tomio.

**Writing – review & editing:** Noriko Morioka, Mutsuko Moriwaki, Jun Tomio, Kiyohide Fushimi, Yasuko Ogata.

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
