## [Decision Letter · Decision Letter 0]

26 Feb 2021

PONE-D-20-29624

Dementia and patient outcomes after hip surgery in older patients: a retrospective observational study using nationwide administrative data in Japan

PLOS ONE

Dear Dr. Morioka,

Thank you for submitting your manuscript to PLOS ONE. After careful consideration, we feel that it has merit but does not fully meet PLOS ONE’s publication criteria as it currently stands. Therefore, we invite you to submit a revised version of the manuscript that addresses the points raised during the review process.

We look forward to receiving your revised manuscript.

Kind regards,

Itamar Ashkenazi

Academic Editor

PLOS ONE

Journal Requirements:

Reviewers' comments:

Reviewer's Responses to Questions

**Comments to the Author**

1. Is the manuscript technically sound, and do the data support the conclusions?

Reviewer #1: Yes

Reviewer #2: Yes

2. Has the statistical analysis been performed appropriately and rigorously? 

Reviewer #1: Yes

Reviewer #2: I Don't Know

3. Have the authors made all data underlying the findings in their manuscript fully available?

Reviewer #1: Yes

Reviewer #2: Yes

4. Is the manuscript presented in an intelligible fashion and written in standard English?

Reviewer #1: Yes

Reviewer #2: Yes

5. Review Comments to the Author

Reviewer #1: The authors did a great job by looking into dementia and its relationship with in-hospital complications after hip replacement surgeries.

The authors used data from a national database, including over 400 hospitals and almost 50 thousand participants.

Through multilevel logistic regressions the authors could assess the nature of the association between dementia and the covariates in a very accurate and statistically appropriate way.

The results shown no association between dementia and in-hospital deaths, in-hospital pneumonia or in-hospital fractures.

Overall, the paper is really well written and I believe it has some relevance to health workers dealing with elderly either with or without dementia.

After reading the manuscript I have only two questions for the authors:

1 - Did the patients have elective or emergency hip replacement surgeries?

Despite the surgery procedure being the same in both situations, the condition that originate the need for the surgery could affect the outcomes, therefore the type of surgery should be disclosed.

2 - What does the “other” type of hospital mean?

I am not familiar with types of hospital in Japan, but it feels odd to have the largest category labelled as “other” in a table designed to explain the types of hospital administration. A definition of what consist this category should be added to the table’s footnote.

Reviewer #2: The study is within the scope of PLOS One, and seems to be conducted in a sound and structured way, although the authors are recommended to read the STROBE or RECORD guidelines and follow them when reporting.

The findings on care quality, measured in patients per nurse is new knowledge to me, and very interesting. Improving the quality of care and outcome after hip fracture is an important task. Still, I feel a bit unsure on the clinical perspective in the way the authors have set up their analysis. It will still be a fact that individuals with dementia faces a higher risk of dying, catch pneumonia or suffer additional fractures. Then, if this is associated with their diagnosis of cognitive impairment or other factors will – in my mind – be of theoretical interest only. It is tempting to agree with the authors in their suggested explanation that it is thanks to an improved dementia care in Japan, but as the authors write this is difficult to prove. A conclusion like “Dementia was not significantly associated with poor patient outcomes among people aged 65 or older who underwent hip surgery, after adjusting for individual and hospital factors” can in worst case lead the reader to believe that patients with dementia do not need any special attention. If dementia per se is not the problem, then the searchlight need to be focused on other risk factors.

The finding “In the present study, people admitted from home or discharged to home were more likely to have longer hospitalizations than those admitted from and discharged to long-term care facilities and other hospitals, regardless of dementia status.” is assumingly the same in many other countries. Persons living “on the edge” in the own home, barely managing independent living, will after a hip fracture need another type of living, at least for a period. Organizing this will take more time than just dismiss a patient back to a previous nursing home place. Here a more international outlook in the Discussion is recommended.

Minor comments:

Understand that only association, not causality, can be studied. The sentence “To investigate whether dementia leads to incidence of adverse events and longer hospital stays” suggests causality.

Japan has a known high rate of non-operatively treated hip fracture cases in international comparisons. How was this group handled in the flow-chart? Assumingly, individuals with severe dementia might be missing in the current analysis, if they are treated mainly non-operatively?

The rate of pneumonia is very low compared to the (historical) literature on hip fracture complications. Is this due to underreporting, better modern care in Japan, less predisposal in Japanese people to get pneumonia than Western world populations?

ADL is mentioned as one of the variables accessible in the data base. Then, in limitations, it is mentioned as an unmeasured confounder. Please, explain.

Refrain from presenting percent with two decimals.

The tables will benefit from a better lay-out for better overview and a neater look. The flowchart needs better wordings, for example individuals or n=xxx not “samples”. Also, report what the reference is in Table 4 (dementia or non-dementia?).

6. PLOS authors have the option to publish the peer review history of their article (what does this mean?). If published, this will include your full peer review and any attached files.

Reviewer #1: **Yes: **Daniel Pozzobon

Reviewer #2: **Yes: **Cecilia Rogmark

---

## [Author Response · Author response to Decision Letter 0]

11 Mar 2021

We sincerely appreciate your careful review and comments, which are very helpful, and we have carefully revised the manuscript accordingly as follows.

Comments to the Author

5. Review Comments to the Author

Reviewer #1: The authors did a great job by looking into dementia and its relationship with in-hospital complications after hip replacement surgeries.

The authors used data from a national database, including over 400 hospitals and almost 50 thousand participants.

Through multilevel logistic regressions the authors could assess the nature of the association between dementia and the covariates in a very accurate and statistically appropriate way.

The results shown no association between dementia and in-hospital deaths, in-hospital pneumonia or in-hospital fractures.

Overall, the paper is really well written and I believe it has some relevance to health workers dealing with elderly either with or without dementia.

>Author’s Response to Reviewer #1

First of all, we sincerely appreciate your careful review and comments, which are very helpful, and we have carefully revised the manuscript accordingly.

After reading the manuscript I have only two questions for the authors:

1 - Did the patients have elective or emergency hip replacement surgeries?

Despite the surgery procedure being the same in both situations, the condition that originate the need for the surgery could affect the outcomes, therefore the type of surgery should be disclosed.

>Author’s Response

Thank you for the comment. Unfortunately, we could not obtain more detail regarding the type of surgery from the database we used. We agree that the type of surgery (scheduled or emergency) could affect patient outcomes. Therefore, we have added this as a limitation.

L363

Second, there may be some unmeasured confounders that might affect patient outcomes (e.g., whether it was a scheduled or emergency surgery and the severity of dementia).

2 - What does the “other” type of hospital mean?

I am not familiar with types of hospital in Japan, but it feels odd to have the largest category labelled as “other” in a table designed to explain the types of hospital administration. A definition of what consist this category should be added to the table’s footnote.

>Author’s Response

Thank you for the comment.　We used the same definition of the type of hospital as used in the Survey of Medical Institutions (https://www.mhlw.go.jp/english/database/db-hss/dl/Definitions_2014.pdf). “Others” included public interest corporations, private university corporations, social welfare corporations, medical co-op, companies, and other corporations. We have now added this description under Methods and as a footnote for Table 2.

L196

We also considered the hospital size (number of beds) and type of established organization (national; public; social; private; and others including public interest corporations, private university corporations, social welfare corporations, medical co-op, companies, and other corporations) as other hospital characteristics.

L242 Footnote of Table 2. Hospital characteristics (N = 404)

a public interest corporations, private university corporations, social welfare corporations, medical co-op, companies, and other corporations.

Reviewer #2: The study is within the scope of PLOS One, and seems to be conducted in a sound and structured way, although the authors are recommended to read the STROBE or RECORD guidelines and follow them when reporting.

>Author’s Response to Reviewer #2

First of all, we sincerely appreciate your careful review and comments, which are very helpful, and we have carefully revised the manuscript accordingly. We confirmed to follow the STROBE guidelines.

The findings on care quality, measured in patients per nurse is new knowledge to me, and very interesting. Improving the quality of care and outcome after hip fracture is an important task. Still, I feel a bit unsure on the clinical perspective in the way the authors have set up their analysis. It will still be a fact that individuals with dementia faces a higher risk of dying, catch pneumonia or suffer additional fractures. Then, if this is associated with their diagnosis of cognitive impairment or other factors will – in my mind – be of theoretical interest only. It is tempting to agree with the authors in their suggested explanation that it is thanks to an improved dementia care in Japan, but as the authors write this is difficult to prove. A conclusion like “Dementia was not significantly associated with poor patient outcomes among people aged 65 or older who underwent hip surgery, after adjusting for individual and hospital factors” can in worst case lead the reader to believe that patients with dementia do not need any special attention. If dementia per se is not the problem, then the searchlight need to be focused on other risk factors.

>Author’s Response

Thank you for the comment. In this study, we focused on whether dementia is associated with poor patient outcomes after hip surgery. The associations between dementia and patient outcomes were not statistically significant in this study, which is inconsistent with the findings of previous studies. However, numerous studies have suggested that people with dementia or cognitive impairment need specialized care during hospitalization. In this study, the overall quality of care for older adults with cognitive impairment in an acute care setting might help reduce the risk of dying, catching pneumonia, or suffering additional fractures. We do not want the reader to believe that patients with dementia do not need any special attention, as you have pointed out. Therefore, we have revised the Conclusion as follows.

L55 Abstract

Conclusions: Although adverse events are more likely to occur in older adults with dementia than in those without dementia after hip surgery, we found no evidence of an association between dementia and adverse events or the length of hospital stay after adjusting for individual social and nursing care environment.

L299 Discussion

We conducted a retrospective observational study with a large sample using nationwide administrative data in Japan to explore the association between dementia and patient outcomes after hip surgery among older adults. Although in-hospital death, in-hospital pneumonia, and in-hospital fracture were more likely to occur in people with dementia than in those without dementia in the present study, they were not significantly associated with dementia after adjusting for potential confounders such as individual social factors, nurse staffing level, and dementia care expertise.

L375 Conclusion

Adverse events were more likely to occur in people aged 65 or older with dementia than in those without dementia after hip surgery. These events were not significantly associated with dementia after adjusting for individual social and nursing care environment. Further studies are necessary to identify factors that mitigate the effect of dementia on poor outcomes.

The finding “In the present study, people admitted from home or discharged to home were more likely to have longer hospitalizations than those admitted from and discharged to long-term care facilities and other hospitals, regardless of dementia status.” is assumingly the same in many other countries. Persons living “on the edge” in the own home, barely managing independent living, will after a hip fracture need another type of living, at least for a period. Organizing this will take more time than just dismiss a patient back to a previous nursing home place. Here a more international outlook in the Discussion is recommended.

>Author’s Response

Thank you for the comment. Previous studies in the Netherlands (reference No. 8) and Canada (reference No. 10) have reported that patients often quickly return to their previous nursing home and therefore have a shorter length of stay compared to those admitted from their own homes. Also, the Canadian study reported that the occurrence of a hip fracture was often associated with transition to long-term care facilities among many community-dwelling older adults. Considering this, coordinating the discharge of patients requiring post-operative care to their homes is time-consuming, as you have pointed out. Therefore, we have added this international perspective in the Discussion as follows.

L326

Past research in other countries also reported that patients often quickly return to their previous nursing home and therefore have a shorter length of stay compared to those admitted from their own homes [8,10].

Minor comments:

Understand that only association, not causality, can be studied. The sentence “To investigate whether dementia leads to incidence of adverse events and longer hospital stays” suggests causality.

>Author’s Response

Thank you for the comment. We revised the sentence as follows.

L33

To investigate whether dementia associated with incidence of adverse events and longer hospital stays

Japan has a known high rate of non-operatively treated hip fracture cases in international comparisons. How was this group handled in the flow-chart? Assumingly, individuals with severe dementia might be missing in the current analysis, if they are treated mainly non-operatively?

>Author’s Response

Thank you for the comment. In this study, we have focused on cases with hip surgery and only included patients who have undergone surgery. As you have pointed out, our analysis did not include individuals with severe dementia who were treated non-operatively. We could compare people with relatively mild dementia or without dementia, but it might involve overlooking the association between dementia and poor patient outcomes. However, we could not obtain the data on the severity of dementia and investigate the above point. We have added this point as a limitation and added a new reference (No. 37).

L363

Second, there may be some unmeasured confounders that might affect patient outcomes (e.g., whether it was a scheduled or emergency surgery and the severity of dementia). Patients with severe dementia are more likely to choose non-operative treatment for hip fractures [37]. In this study, we focused on cases with hip surgery, which might have involved those with relatively mild dementia.

The rate of pneumonia is very low compared to the (historical) literature on hip fracture complications. Is this due to underreporting, better modern care in Japan, less predisposal in Japan nese people to get pneumonia than Western world populations?

>Author’s Response

Thank you for the comment. In our data, the prevalences of pneumonia among patients with and without dementia were 0.15% and 0.07%, respectively, which were relatively low compared to those reported in previous studies both in other countries and in Japan. For example, Mosk et al. (reference no. 8) reported that the prevalence of postoperative pneumonia was 3.18% among 566 patients aged >=70 years with an isolated hip fracture in a hospital in the Netherlands. The Japanese Orthopaedic Association conducted a registry survey from 2009 to 2014 and reported the prevalence of pneumonia among patients with hip fracture was 5.7% (15/3007patients) (https://doi.org/10.1016/j.jos.2017.06.003). 

The relatively low prevalence of pneumonia in our study might be explained by the characteristics of the database and the better modern nursing care in the acute care setting. Regarding the DPC database that we used in this study, the validation study reported that the sensitivity of diagnoses was not high and varied across conditions (reference no. 22). Also, this database is linked with a diagnosis-related payment system, and the diagnosis was disclosed only for those who opted for treatment. The low prevalence of pneumonia might be owing to such under-reporting. However, those reasons were the same between patients with and without dementia patients and those didn’t affect our findings. 

Regarding the nursing care environment, we selected hospitals that had an average of > 200 in-patients per day in order to adjust for the effect of surgeon procedure volumes, in which nursing care environment might be better than those in relatively small hospitals as we described in limitation. 

We added the explanation in the limitation as follows.

L370

Third, for the database used in this study, the validation study reported that the sensitivity of diagnoses was not high and varied across conditions [22]. The prevalence of non-critical adverse events, such as mild pneumonia, might have been underestimated. This, however, would not affect our findings since underestimations could occur with or without dementia.

ADL is mentioned as one of the variables accessible in the data base. Then, in limitations, it is mentioned as an unmeasured confounder. Please, explain.

>Author’s Response

Thank you for the comment. Although our dataset included ADL, about one-fourth ADL data were missing and had to be excluded from the analysis. We have added an explanation for this as follows.

L115

In this study, we could not include ADL data in the analysis because almost one-fourth of such data were missing.

L367

The ADL of patients were also related to the length of hospital stay [8,38]. Although our dataset included ADL, some ADL data were missing and thus had to be excluded from the analysis.

Refrain from presenting percent with two decimals.

The tables will benefit from a better lay-out for better overview and a neater look. The flowchart needs better wordings, for example individuals or n=xxx not “samples”. Also, report what the reference is in Table 4 (dementia or non-dementia?).

>Author’s Response

Thank you for the suggestions and comments. We have replaced “samples” with “individuals” in the Fig. S1. We also revised the tile of Table 4 as follows. We agree with your suggestion that the tables will be decipherable if we refrain from presenting percentage with two decimals. However, the figures of the prevalence of adverse events were low and differed for those with and without dementia in this study. We would appreciate it if you would allow us to use the percentage with two decimals.

L268 

Table 4. Results of multilevel logistic regression analysis for in-hospital death, in-hospital pneumonia, and in-hospital fracture among people with and without dementia (n = 48,797)

S1 Fig.

samples -> individuals

---

## [Editor Report · Decision Letter 1]

17 Mar 2021

Dementia and patient outcomes after hip surgery in older patients: a retrospective observational study using nationwide administrative data in Japan

PONE-D-20-29624R1

Dear Dr. Morioka,

We’re pleased to inform you that your manuscript has been judged scientifically suitable for publication and will be formally accepted for publication once it meets all outstanding technical requirements.

Kind regards,

Itamar Ashkenazi

Academic Editor

PLOS ONE
---

## [Editor Report · Acceptance letter]

13 Apr 2021

PONE-D-20-29624R1 

Dementia and patient outcomes after hip surgery in older patients: a retrospective observational study using nationwide administrative data in Japan 

Dear Dr. Morioka:

I'm pleased to inform you that your manuscript has been deemed suitable for publication in PLOS ONE. Congratulations! Your manuscript is now with our production department. 

Kind regards, 

on behalf of

Dr. Itamar Ashkenazi 

Academic Editor

PLOS ONE